# Toward integrating clinical and non-clinical associates of suicidality to inform potential intervention points among youth in Nairobi metropolitan, Kenya

suicidality; severity; intensity; environment; adolescents

**Corresponding author:**
David Ndetei;
Email: dmndetei@amhf.or.ke

David Ndetei[1,2] ⬤, Danuta Wasserman[3], Victoria Mutiso[1], Kamaldeep Bhui[4] ⬤, Jenelle Shanley[5], Christine Musyimi[1], Samantha Winter[6], Pascalyne Nyamai[1], Samuel Walusaka[1], Veronica Onyango[1], Eric Jeremiah[1], Tom Lee Osborn[7] ⬤, Monica Swahn[8], Andre Sourander[9] and Daniel Mamah[10]

[1]Africa Institute of Mental and Brain Health, Nairobi, Kenya; [2]Psychiatry, University of Nairobi, Kenya; [3]Karolinska Institute, Sweden; [4]Psychiatry, University of Oxford, UK; [5]Pacific University, USA; [6]Columbia University School of Social Work, USA; [7]Shamiri Institute Inc, Kenya; [8]Kennesaw State University College of Humanities and Social Sciences, USA; [9]University of Turku Research Centre of Applied and Preventive Cardiovascular Med, Finland and [10]Washington University in St Louis, USA

## Abstract

Suicide is a significant global public health concern, particularly among adolescents, with substantial implications for economies, societies and individuals' mental well-being. Understanding its patterns and intention and psychosocial determinants in a given context can suggest potential intervention points. This population-based cross-sectional study aimed to document suicidal ideas, behaviors and intensity among youths aged 14 to 25 in the Nairobi metropolitan area and associated socio-economic position, demographic indicators and potential intervention points. A diverse sample of 1,972 participants was recruited from urban and peri-urban settings within the Nairobi metropolitan area. Data analysis included descriptive statistics, chi-square tests and logistic regression. Our findings confirm a high prevalence of suicidal ideas and behavior in the youth (19.9% and 3.6%, respectively), with very few significant differences between the urban and peri-urban areas. The severity of suicidal ideation and behavior reported methods and reasons, and the socio-demographic profile of participants, point to multiple potential intervention targets. These findings ought to be used to design, manage and evaluate suicide prevention programs.

## Impact statement

The study presents important findings on youth suicide and risk factors in the Nairobi metropolitan area, thereby addressing an essential knowledge gap in low- and middle-income countries. In particular, this study indicates that one-fifth of young people have suicidal thoughts; some have attempted suicide. Social demographics factors like education, marital status and living standards influence behavior, highlighting the need for improved early detection and prevention in developing nations. Furthermore, this study provides evidence for combined clinical intervention with necessary consideration for wide-based environmental, economic, psychosocial and demographic indicators for inclusive interventions. Overall, this study contributes to suicide prevention efforts among youth aged between 14 and 25 years within the Nairobi Metropolitan area by determining the prevalence, the level of severity and the intensity of suicidal ideation differently by age and gender within the target community. By analyzing both clinical and socio-environmental factors such as schooling status, prior psychiatric history and socio-economic context, the findings suggest multiple key areas where intervention and support systems can be targeted. The policy implications and mental health programs can be informed by these findings to help intervene with youth at risk within the school and community, along with determining them.





## Introduction

Globally, suicide is a significant public mental health concern with substantial economic, social and psychological implications. Most suicides occur among youth in low- and middle-income countries (LMICs), which account for most of the world's population (WHO, 2024). These trends

underscore the need for continued research on risk factors – both clinical and non-clinical factors for suicidality, particularly in LMIC settings.

Suicidality refers to the spectrum of thoughts and behaviors related to suicide. These include suicidal ideation (thinking about or planning suicide), suicide plans and suicide attempts (engaging in potentially self-injurious behavior with at least some intent to die). While these elements are interconnected and often studied together, they represent distinct stages within a continuum of suicide risk. Various factors have been associated with suicidality, including mental health disorders such as depression (Ndetei et al., 2010, 2022), financial difficulties or low economic status (Musyimi et al., 2020), family instability or stressors, (Mutwiri et al., 2023) and experiences of bullying, whether in person or online (Mutiso et al., 2019).

In high-income countries (HIC), suicidal behaviors and suicidal thoughts among young people are often studied as part of the same risk continuum. However, they represent distinct aspects of suicidality, and their prevalence varies across studies (Ivey-Stephenson et al., 2022; Strandheim et al., 2014). In both HIC and LMICs, inconsistencies exist regarding prevalence, patterns, study participants and environmental factors (Núñez et al., 2019).

Several studies in Africa, including Uganda, Ethiopia and Ghana, report varying prevalence of suicidal ideas and behaviors (Bukuluki et al., 2021; Culbreth et al., 2018; Quarshie et al., 2019). In Kenya, a study conducted at a youth center within a tertiary referral hospital that serves primarily youths with mental health needs found that 82% showed suicidal behaviors, with ages 16–18 at higher risk than 19–22 (Khasakhala et al., 2013). Another Kenyan study among students aged 15 and older investigating suicidal ideation during the previous 2 weeks reported a prevalence of 22.6% (Ndetei et al., 2022).

This study aimed to document clinical severity and intensity of suicidality to identify the underexplored entry points for intervention in the study context.

A major knowledge gap is the lack of multi-faceted and integrated documentation of various clinical features (severity and intensity) and environmental non-clinical factors (economic and demographic risk factors) that can inform integrated, inclusive intervention.

Generally, the study aimed to address the lack of integrated documentation on suicidality by examining clinical and non-clinical factors associated with suicidality among youth in the Nairobi metropolitan area. Specifically, we aimed to (1) investigate the patterns of suicidality among youth living in the Nairobi metropolitan, (2) examine clinical features (ideation, behavior, attempts, severity and intensity) and their relevance to inform potential interventions and (3) investigate socio-demographic and economic indicators as environmental factors associated with suicidality.

## Methodology

### Study design and setting

This was a population-based cross-sectional study examining suicidality among Nairobi metropolitan youth, designed to explore integrated clinical and non-clinical features of suicidality that can inform intervention points. Targeting Nairobi's poorest neighborhoods ensured a diverse and representative sample. Specifically, two informal settlement areas located close to wealthier neighborhoods were chosen to capture a range of youth experiences within low-resource settings.

### Study participants

The study included youths aged 14 to 25 residing within the designated neighborhoods. Recruitment was done in educational institutions (colleges and vocational training centers) during sessions and in the broader community during school closure periods. This approach ensured inclusion of both in-school (those attending classes) and out-of-school youth (including those on break, graduates or school dropouts), capturing diverse lived experiences and risk exposures. Out-of-school youth may experience different psychosocial stressors such as unemployment, early marriage or domestic responsibilities, which informed our inclusive sampling designed to improve the generalizability of findings and inform more equitable interventions. Exclusion criteria encompassed individuals outside the specified age range, those unable to comprehend the questionnaire, including informed consent due to factors such as intoxication or illiteracy, and those unwilling to participate.

### Procedures

#### Research assistant selection and training
Twelve research assistants (RAs) with prior experience underwent a 2-day comprehensive in-person training involving data collection techniques and protocols, including administration of the Columbia Suicide Severity Rating Scale (C-SSRS) and other study tools.

#### Recruitment and resulting sample
All data were collected in the daytime from September 21, 2022, to December 15, 2022, based on a schedule that included a community sensitization phase. This involved engaging existing local administrators, including chiefs, through community opinion leaders (village elders) to raise awareness about the study, explain its purpose and encourage participation. Study sites were selected in consultation with local administrators to ensure representation from economically disadvantaged urban and peri-urban areas while minimizing potential political influence. We involved the County Commissioners in Kiambu and Nairobi counties for approvals. All eligible youth, including marginalized groups such as school dropouts, youth from low socioeconomic backgrounds and those married early, who may otherwise have been underrepresented, were encouraged to participate. This approach ensured representation across different socioeconomic backgrounds, engagement with both high-risk populations and general youth groups, improving the relevance of our findings and minimizing selection bias.

A total of 1,972 participants (both adults and minors) participated in the study after providing written consent and assent. None of the participants declined to participate (consistent with our previous studies, where the response rate has been nearly 100% (Ndetei et al., 2010)).

#### Data collection
Participants were randomized into 12 groups of up to 25 participants, each managed by an RA, solely for logistical purposes and to streamline the data collection process. Each participant completed the self-administered questionnaires (socio-demographic and economic indicators, and the C-SSRS) individually in a private setting, similar to a national examination. Discussions or disclosure of the responses were not allowed to ensure confidentiality and prevent stigma. RAs explained the study, obtained written consent/assent, verified age eligibility and ensured privacy by preventing discussion and providing clarification when needed.

## Tools

### Socio-demographic profile

A self-reported socio-demographic questionnaire was used to gather information on age, gender, marital status, religion, birth position, level of education, employment status, primary source of income, place of abode and whether they were sharing the living space(s).

### Wealth index

The wealth index self-report questionnaire, based on the World Bank recommendation for LMICs, was used to record household items (source of water, the type of flooring, the type of toilet and the method used for cooking) that estimate economic status by creating a wealth index (Smits & Steendijk, 2015). The wealth index is categorized into five quintiles, with Quintile 1 representing the lowest level of wealth and Quintile 5 the highest level. The individual items of these economic indicators, their scores and the compilation of the wealth index have previously been described (Ndetei et al., 2022).

### The Columbia Suicide Severity Rating Scale (C-SSRS)

The self-reported full version of the C-SSRS assessed both lifetime and recent suicidal ideation (past 1 month) and behavior (past 3 months) (Columbia Lighthouse Project, 2016; Posner et al., 2008). Participants took approximately 5 min to complete it independently after administration.

The C-SSRS in our study included:

Section A: Suicidal ideation severity (five items: wish to be dead, nonspecific thoughts, ideation without plan, with intent and with plan),

Section B: Ideation intensity (five items: frequency, duration, controllability, deterrents and reasons),

Section C: Suicidal behavior (12 items): including items on actual attempts, interrupted, aborted attempts and preparatory behaviors.

Sections D–F: Activating events, treatment history and protective factors.

The C-SSRS, originally designed to differentiate suicidal thoughts and behaviors, has been widely translated and used globally. Its reliability and validity in predicting suicide risk are well established across different populations (Austria-Corrales et al., 2023; Schwartzman et al., 2023; Wilson, 2017). The tool has been validated in various LMIC contexts, including in Lebanon, South Africa and Mozambique, demonstrating the scale's relevance and reliability (Lovero et al., 2024; Stockton et al., 2024; Zakhour et al., 2021). The C-SSRS, through standardized definitions and structured questions, has significantly improved the accuracy of suicide risk assessment (Yershova et al., 2016). Moreover, its adoption has influenced suicide research, clinical practice and public health surveillance globally (Posner et al., 2014).

Although C-SSRS is primarily designed as a categorical tool, we calculated Cronbach's alpha to examine the internal consistency of the item responses in our sample. Results indicated strong reliability: Suicidal ideation: $\alpha = .821$ (lifetime) and .801 (recent). Suicidal behavior: $\alpha = .900$ (lifetime) and .879 (recent).

This analysis was done to explore the consistency of responses and not to imply the use of a composite score. Similar approaches have been reported in previous research (Zakhour et al., 2021). Treatment satisfaction was assessed using a binary self-report item asking whether participants were satisfied or dissatisfied with previous psychiatric care.

The full list of items used is available in Supplementary File 1.

## Statistical analysis

### Coding

The C-SSRS is a self-administered tool that measures suicide ideation and behavior. The first subscale (suicidal ideation) assesses five levels of ideation severity, ranging from 1 (*wish to be dead*) to 5 (*suicidal intent with plan*). Youth participants who denied ideation (no response) received a zero, and those who had ideation (yes response) received 1. The second subscale, the intensity scale, is comprised of five items (i.e., frequency, duration, controllability, deterrents and reasons for ideation), each rated on an ordinal scale (total scores ranging from 2 to 25). These five items are completed only with adolescents who endorse at least one of the severity items. Those without any suicidal ideation are given a scale score of 0 on intensity. The behavior scale investigates actual attempts, having eight questions with no and yes responses, interrupted, aborted or self-interrupted attempts, preparatory acts or behavior and self-injurious behavior without suicide intent. The composite variables (ideation and behavior) were computed as binary variables, where individuals who endorsed any level of ideation or behavior were coded as '1' (Yes), and those who did not endorse any ideation or behavior were coded as '0' (No). This approach ensured that the analysis focused on the presence versus absence of suicidality.

### Data processing and analysis

Data cleaning and analysis used IBM SPSS version 25, with descriptive statistics and frequency distributions for suicide patterns and risk factors. The results are presented using both tabular and narrative descriptive summaries. List-wise deletion of cases was applied in the multivariable analysis since most variables had less than 5% of the missing data. The C-SSRS items were analyzed as categorical variables, focusing on the presence or absence of suicidal ideation (past month) and suicidal behavior (past 3 months). Chi-square tests and Fisher's exact tests were carried out to examine the group differences (urban and peri-urban) and the relationship between suicide ideations and behavior (actual attempts and preparatory act) with social demographics. Logistic regression models were conducted on variables with a *p*-value of less than <.05 to investigate predictors of suicidal ideations, attempts and preparatory plans or behavior. Variables such as gender, age, marital status, education, employment status, religion, living situations and economic status were included in the regression model based on their known associations with suicidal ideations and behavior in prior literature (Almeida et al., 2012; Ndetei et al., 2022). This approach ensured that our model captured both statistically significant associations and well-established risk factors in suicidality research. Adjusted odds ratios (AOR) with 95% confidence intervals (CI) were calculated to assess the strength and significance of the association. All tests were two-sided, and statistical significance was set at *p* < .05. Outcome variables were suicidal ideation and suicidal behavior subscales. Additionally, reasons for suicidal ideation were a multiple-choice item with an "Other (specify)" option. Responses were analyzed quantitatively, while write-in responses were categorized thematically for descriptive purposes.

## Ethics

The authors assert that all procedures contributing to this work comply with the ethical standards of the relevant national and institutional committees on human experimentation and with the

Helsinki Declaration of 1975, as revised in 2008. All procedures involving human subjects/patients were approved by the Nairobi Hospital Ethics Research Committee (approval no. TNH-ERC/DMSR/ERP/022/22). The study obtained licensing from the National Commission for Science, Technology and Innovation (NACOSTI) license number NACOSTI/P/22/18097. Permissions were obtained from county offices and colleges. Before data collection, adults provided informed consent while minors under 18 provided assent accompanied by a parent or legal guardian who provided consent in line with ethics committee guidance. Consent procedures were conducted in private spaces to ensure voluntary participation and confidentiality. Participants determined to be at risk of suicide were referred to public psychiatric facilities for evaluation on underlying psychiatric conditions and care. Participants were given verbal study explanations, could ask questions and could withdraw at any time without penalty.

## Results

### Socio-demographic and economic characteristics

A total of 1,972 participants (both adults and minors) participated in the study. More than half of the participants were female (55%), majority identified as Christians (87.2%). Over half of the participants (52.4%) had completed secondary school, majority living in urban areas (57%). The majority were unemployed (77.9%) and had never been married (60.4%) since they were mostly students. The majority of participants (74.5%) fell within the second and third wealth index categories, indicating predominantly lower to middle economic status. There were significant differences between peri-urban and urban populations ($p < 0.001$). Peri-urban participants were younger ($M = 19.85$ vs. 20.55), more likely protestant (54.9% vs. 42.8%) and had higher tertiary education (95.8% vs. 25.7%). See Table 1.

**Table 1.** Social demographics of participants by location

| Variables | Categories | Total (N = 1972) | Peri-urban 168 (8.5%) | Urban 1804 (91.5%) | F | t | p-value |
|---|---|---|---|---|---|---|---|
| | | 20.49 ± 2.631 (14,25) | 19.85 ± 1.447 (17,24) | 20.55 ± 2.699 (14,25) | 73.666 | −5.362 | p < 0.001 |
| Age | Mean ± SD (Min,Max) | N (%) | N (%) | N (%) | Chi-square | Df | p-value |
| Gender | Female | 1,069 (55.0%) | 95 (57.6%) | 974 (54.7%) | 0.498 | 1 | 0.48 |
| | Male | 876 (45.0%) | 70 (42.4%) | 806 (45.3%) | | | |
| Religion | Protestant | 816 (43.9%) | 90 (54.9%) | 726 (42.8%) | 24.284 | 3 | **<0.001** |
| | Catholic | 805 (43.3%) | 54 (32.9%) | 751 (44.3%) | | | |
| | Muslim | 103 (5.5%) | 17 (10.4%) | 86 (5.1%) | | | |
| | Other | 135 (7.3%) | 3 (1.8%) | 132 (7.8%) | | | |
| Level of education | Primary | 179 (9.1%) | 1 (0.6%) | 178 (9.9%) | 349.68 | 3 | **<0.001** |
| | Secondary | 1,031 (52.4%) | 5 (3.0%) | 1,026 (57.0%) | | | |
| | Tertiary | 623 (31.7%) | 161 (95.8%) | 462 (25.7%) | | | |
| | University | 135 (6.9%) | 1 (0.6%) | 134 (7.4%) | | | |
| Current marital status | Never married | 1,182 (60.4%) | 130 (78.3%) | 1,052 (58.7%) | 35.823 | 6 | **< 0.001** |
| | In a relationship | 428 (21.9%) | 27 (16.3%) | 401 (22.4%) | | | |
| | Married/cohabiting | 225 (11.5%) | 1 (0.6%) | 224 (12.5%) | | | |
| | Separated (not divorced) | 39 (2.0%) | 1 (0.6%) | 38 (2.1%) | | | |
| | Divorced | 16 (0.8%) | 0 (0.0%) | 16 (0.9%) | | | |
| | Widowed | 8 (0.4%) | 0 (0.0%) | 8 (0.4%) | | | |
| | Other | 59 (3.0%) | 7 (4.2%) | 52 (2.9%) | | | |
| Wealth index | 1. Lowest | 284 (15.3%) | 8 (5.1%) | 276 (16.2%) | 78.124 | 4 | **<0.001** |
| | 2 | 703 (37.8%) | 43 (27.4%) | 660 (38.8%) | | | |
| | 3 | 681 (36.7%) | 60 (38.2%) | 621 (36.5%) | | | |
| | 4 | 166 (8.9%) | 41 (26.1%) | 125 (7.3%) | | | |
| | 5.Highest | 24 (1.3%) | 5 (3.2%) | 19 (1.1%) | | | |
| Current employment status | Unemployed | 1,531 (77.9%) | 156 (92.9%) | 1,375 (76.5%) | 24.753 | 3 | **<0.001** |
| | Volunteering | 225 (11.5%) | 9 (5.4%) | 216 (12.0%) | | | |
| | Part time employment | 75 (3.8%) | 1 (0.6%) | 74 (4.1%) | | | |
| | Full time employment | 134 (6.8%) | 2 (1.2%) | 132 (7.3%) | | | |

*Note:* Df, Degree of freedom; *t*, *t* value. Significant *p*-values are indicated in bold.

### The different suicidal items

From Table 2, the prevalence of suicidal ideation and behaviors among participants, 19.9% reported lifetime suicidal thoughts, 16.7% had intent to act and 13.8% wished to be dead. Suicide attempts occurred in 5.5% lifetime and 3.6% recently. Non-suicidal self-injury has affected 4.9% recently. Lifetime lethal actions were 8.1%, with 6.7% attempting suicide and 7.9% believing their actions could be fatal. Over the past 3 months, reported behaviors generally decreased compared to lifetime experiences.

*Suicidal items distribution across peri-urban and urban*: Lifetime suicide attempts showed significant differences, $\chi^2$ (1) = 4.142, $p$ = .042, with peri-urban areas having a higher prevalence (10.5%) than urban areas (6.3%). Furthermore, peri-urban areas had a lower rate of self-harming behavior without suicidal intent (2.0%) compared to urban areas (5.8%), $\chi^2$ (1) = 3.945, $p$ = .047. For recent behaviors, significantly higher urban participants (4.3%) stopped themselves before attempting suicide compared to peri-urban participants (0.7%), $\chi^2$ (1) = 4.091, $p$ = .043.

### Severity of suicidal ideation levels prevalence (lifetime and past 1 month) (narrative results)

During the lifetime, approximately 45% ($n$ = 54) of participants reported thoughts of wishing to die, followed by Levels 2 (15.0%, $n$ = 18), 3 (14.2%, $n$ = 17) and 4 (14.2%, $n$ = 17), and lastly, 5 (11.7%, $n$ = 14) had active suicidal ideation with a specific plan and intent. In recent cases, about 47.4% ($n$ = 45) of participants wished to die, 13.7% had active ideation thoughts and a plan, while 11.6% had active thoughts with intent but without a specific plan, followed by Levels 2 (13.7%, $n$ = 13), 3 (13.7%, $n$ = 13), 4 (11.6%, $n$ = 11) and 5 (13.7%, $n$ = 13).

### The intensity of ideation; frequency, duration, controllability, deterrents and reasons for ideation

Table 3 highlights suicidal ideation patterns among 319 respondents. Most experienced thoughts less than once a week (45.5%), with some reporting weekly to daily frequencies. Duration varied, with 32.2% having thoughts lasting 1–4 h and 15.9% having fleeting thoughts. Controllability differed, with 8.6% unable to control thoughts and 40.6% easily controlling them. Deterrents were reported by 35.1%. About 36.6% and 21.4% cited the desire and complete desire, respectively, to end or stop pain as the main reason for ideation.

### Reasons for suicidal ideations (drawn from open-ended responses among participants who endorsed such thoughts) (narrative)

Participants with suicidal ideations attributed the suicidal ideation to various reasons including depression (43%), financial hardship (24%), familial stressors and/or violence (17%), loneliness and feeling discriminated (13%), sickness (1%) and other stressors including work life, self-esteem and school performance (2%).

### Frequency of recent activating events (narrative)

About 51.8% of respondents acknowledged current or pending isolation or feeling alone as activating events of suicidal ideation and behavior, while 29.2% of respondents reported experiencing a recent loss or other significant negative event.

### The prevalence of the treatment history among the surveyed population (narrative)

Of the total sample ($N$ = 1,972), $n$ = 23.4% of the participants with valid responses (181/774) reported previous psychiatric diagnoses and treatments, while 18.8% (140/773) reported feeling hopeless or dissatisfied with their treatment. Additionally, 19.8% (148/746) were noncompliant with treatment, and 33.3% (268/804) were not receiving treatment at all.

### Recent protective factors (narrative)

Responsibility toward family or living with family (61.5%), identifying reasons for living (61.2%) and supportive social networks or family (60.2%) are the most prevalent protective factors. Engagement in work or school (58.9%) was a protective factor. Of the participants who responded to the question "Fear of death or dying due to pain and suffering," (53.6%) agreed to it as the recent protective factor. Similarly, a belief that suicide is immoral, coupled with high spirituality, was reported by 54.80% of respondents.

### Suicidal behavior and types of suicide attempts (narrative)

In total, 15% of participants reported a lifetime suicide attempt, 8.6% aborted attempts, 10.6% non-suicidal self-injurious (NSSI), 8.2% interrupted attempts and 7.3% preparatory acts. In the past 3 months, actual attempts decreased to 9.5%, while other forms, including NSSI (11.4%), remained similar.

Most participants stated that engaging in risky behavior to provoke a road traffic accident (28%), such as speeding on a motorcycle or recklessly crossing roads, as the common suicide method. Other methods, in order of frequency, included self-choking (self-sabotage and stealing to be caught) (15%), hanging, burning oneself, playing with fire, unsafe abortion, ingesting poison, attempting drowning, jumping from a height, self-cutting, self-sabotage through theft, engaging in fights and overdosing.

### Associations between demographic and economic position correlates and suicide ideations and behavior (attempts and preparatory acts)

Females and respondents who were previously married (divorced and separated) were significantly associated with higher suicidal ideations based on chi-square test. The wealth index showed significance for lifetime suicidal attempts ($\chi^2$ = 12.194, $p$ = 0.016) and preparatory acts ($\chi^2$ = 19.377, $p$ = 0.001), while lower age groups were highly associated with aborted or self-interrupted suicide. Primary and secondary education, as the highest levels of education, were associated with high preparatory acts. See Table 4.

### Social demographics as predictors of suicide ideations and behavior

Males had a lower likelihood of suicidal ideation (COR = 0.76, 95% CI: 0.59–0.98) and interrupted attempts (COR = 0.63, 95% CI: 0.40–0.998), compared to females. Age was significantly associated with actual attempts (AOR = 1.07, 95% CI: 1.01–1.13). Lower wealth index increased the risk of preparatory acts. Previously married individuals faced higher suicide risks. Higher education levels reduced suicidal behaviors. Living in hosted accommodations lowered suicide attempts and ideation compared to family-owned residences. See Table 5.

**Table 2.** Prevalence of lifetime and recent suicide ideation and behavior across peri-urban and urban populations

| | Lifetime | | | | | Recent (Ideation; past 1 month, behavior; past 3 months) | | | | |
| | Yes count (*n*) (%) | | | | | Yes count (*n*) (%) | | | | |
| Suicide items | Total | Peri-urban | Urban | Chi-square (df) | *p*-value | Total | Peri-urban | Urban | $\chi^2$ (df)/ Fisher's exact | *p*-value |
|---|---|---|---|---|---|---|---|---|---|---|
| 1. Have you wished you were dead or wished you could go to sleep and not wake up? | 269 (13.8%) | 27 (16.3%) | 242 (13.6%) | $\chi^2(1) = 0.914$ | *p* = .339 | 119 (7.0%) | 12 (7.7%) | 107 (6.9%) | $\chi^2(1) = 0.158$ | *p* = .691 |
| 2. Have you actually had any thoughts of killing yourself? | 225 (11.7%) | 25 (15.2%) | 200 (11.4%) | $\chi^2(1) = 2.048$ | *p* = .152 | 109 (6.5%) | 12 (7.7%) | 97 (6.3%) | $\chi^2(1) = 0.437$ | *p* = .509 |
| 3. Have you been thinking about how you might do this? | 183 (19.9%) | 14 (24.6%) | 169 (19.6%) | $\chi^2(1) = 0.815$ | *p* = .367 | 99 (12.3%) | 9 (16.1%) | 90 (12.0%) | $\chi^2(1) = 0.780$ | *p* = .377 |
| 4. Have you had these thoughts and had some intention of acting on them? | 145 (16.7%) | 11 (20.8%) | 134 (16.5%) | $\chi^2(1) = 0.652$ | *p* = .420 | 82 (10.9%) | 6 (11.5%) | 76 (10.9%) | $\chi^2(1) = 0.020$ | *p* = .888 |
| 5. Have you started to work out or worked out the details of how to kill yourself? Do you intend to carry out this plan? | 96 (11.6%) | 4 (7.8%) | 92 (11.8%) | $\chi^2(1) = 0.736$ | *p* = .391 | 52 (7.1%) | 4 (8.2%) | 48 (7.1%) | Fisher's exact | *p* = .772 |
| 6. Have you made a suicide attempt in your lifetime? | 126 (6.7%) | 17 (10.5%) | 109 (6.3%) | $\chi^2(1) = 4.142$ | ***p* = .042*** | 61 (3.6%) | 3 (2.0%) | 58 (3.8%) | $\chi^2(1) = 1.154$ | *p* = .283 |
| 7. Have you done anything to harm yourself in your lifetime? | 126 (7.6%) | 11 (6.8%) | 115 (7.7%) | $\chi^2(1) = 0.155$ | *p* = .694 | 56 (3.8%) | 4 (2.7%) | 52 (3.9%) | $\chi^2(1) = 0.464$ | *p* = .496 |
| 8. Have you done anything dangerous where you could have died in your lifetime? | 134 (8.1%) | 15 (9.4%) | 119 (8.0%) | $\chi^2(1) = 0.393$ | *p* = .530 | 64 (4.2%) | 8 (5.5%) | 56 (4.1%) | $\chi^2(1) = 0.654$ | *p* = .419 |
| 9. Did you do this as a way to end your life? | 81 (5.1%) | 6 (4.0%) | 75 (5.2%) | $\chi^2(1) = 0.443$ | *p* = .506 | 42 (3.0%) | 2 (1.4%) | 40 (3.2%) | Fisher's exact | *p* = .308 |
| 10. Did you want to die (even a little) when you did this? | 75 (4.7%) | 8 (5.3%) | 67 (4.7%) | $\chi^2(1) = 0.100$ | *p* = .751 | 46 (3.4%) | 2 (1.4%) | 44 (3.6%) | Fisher's exact | *p* = .315 |
| 11. Were you trying to end your life when you did this? | 79 (5.1%) | 7 (4.6%) | 72 (5.1%) | $\chi^2(1) = 0.061$ | *p* = .805 | 40 (2.9%) | 3 (2.2%) | 37 (3.0%) | Fisher's exact | *p* = .791 |
| 12. Did you think it was possible you could have died from this? | 123 (7.9%) | 12 (8.0%) | 111 (7.9%) | $\chi^2(1) = 0.003$ | *p* = .958 | 62 (4.6%) | 6 (4.4%) | 56 (4.6%) | $\chi^2(1) = 0.006$ | *p* = .939 |
| 13. Did you do it purely for other reasons/without ANY intention of killing yourself (like to relieve stress, feel better, get sympathy or get something else to happen)? (self-injurious behavior without suicidal intent) | 84 (5.4%) | 3 (2.0%) | 81 (5.8%) | $\chi^2(1) = 3.945$ | ***p* = .047*** | 38 (2.7%) | 5 (3.7%) | 33 (2.6%) | Fisher's exact | *p* = .413 |
| 14. Have you engaged in non-suicidal self-injurious behavior in your lifetime? | 135 (7.9%) | 14 (9.5%) | 121 (7.7%) | $\chi^2(1) = 0.563$ | *p* = .453 | 74 (4.9%) | 7 (5.2%) | 67 (4.9%) | $\chi^2(1) = 0.036$ | *p* = .849 |
| 15. Has there been a time when you started to do something to end your life but someone or something stopped you before you actually did anything? | 104 (6.1%) | 11 (7.5%) | 93 (5.9%) | $\chi^2(1) = 0.593$ | *p* = .441 | 56 (3.6%) | 4 (2.9%) | 52 (3.6%) | Fisher's exact | *p* = .813 |
| 16. Has there been a time when you started to do something to try to end your life but you stopped yourself before you actually did anything? | 109 (6.4%) | 5 (3.5%) | 104 (6.6%) | $\chi^2(1) = 2.205$ | *p* = .138 | 62 (4.0%) | 1 (0.7%) | 61 (4.3%) | $\chi^2(1) = 4.091$ | ***p* = .043*** |
| 17. Have you taken any steps toward making a suicide attempt or preparing to kill yourself? | 93 (5.5%) | 6 (4.2%) | 87 (5.6%) | $\chi^2(1) = 0.520$ | *p* = .471 | 50 (3.3%) | 1 (0.8%) | 49 (3.5%) | Fisher's exact | *p* = .120 |

*Note:* Bold values are significant.

**Table 3.** Frequency of various aspects of intensity of ideation

| Aspects of ideation | Category | Frequency (n = 319) | n (%) |
|---|---|---|---|
| *1. Frequency*<br>How many times have you had these thoughts? | Less than once a week | 145 | 45.5 |
| | Once a week | 12 | 3.8 |
| | 2–5 times in week | 91 | 28.5 |
| | Daily or almost daily | 34 | 10.7 |
| | Many times each day | 37 | 11.6 |
| *2. Duration*<br>When you have the thoughts, how long do they last? | Fleeting – few seconds or minutes | 50 | 15.9 |
| | Less than 1 h/some of the time | 91 | 29.0 |
| | 1–4 h/a lot of time | 101 | 32.2 |
| | 4–8 h/most of day | 27 | 8.6% |
| | More than 8 h/persistent or continuous | 45 | 14.3 |
| *3. Controllability*<br>Could/can you stop thinking about killing yourself or wanting to die if you want to? | Does not attempt to control thoughts | 19 | 6.1 |
| | Easily able to control thoughts | 126 | 40.6 |
| | Can control thoughts with little difficulty | 63 | 20.3 |
| | Can control thoughts with some difficulty | 36 | 11.6 |
| | Can control thoughts with a lot of difficulty | 40 | 12.9 |
| | Unable to control thoughts | 26 | 8.4 |
| *4. Deterrents*<br>Are there things – anyone or anything (e.g., family, religion and pain of death) – that stopped you from wanting to die or acting on thoughts of committing suicide? | Does not apply | 75 | 25.1 |
| | Deterrents definitely stopped you from attempting suicide | 105 | 35.1 |
| | Deterrents probably stopped you | 51 | 17.1 |
| | Uncertain that deterrents stopped you | 31 | 10.4 |
| | Deterrents most likely did not stop you | 24 | 8.0 |
| | Deterrents definitely did not stop you | 13 | 4.3 |
| *5. Reasons for ideation*<br>What sort of reasons did you have for thinking about wanting to die or killing yourself? Was it to end the pain or stop the way you were feeling (in other words you could not go on living with this pain or how you were feeling) or was it to get attention | Does not apply | 47 | 15.9 |
| | Completely to get attention, revenge or a reaction from others | 27 | 9.2 |
| | Mostly to get attention, revenge or a reaction from others | 16 | 5.4 |
| | Equally to get attention, revenge or a reaction from others and to end/stop the pain | 34 | 11.5 |
| | Mostly to end or stop the pain (you could not go on living with the pain or how you were feeling) | 108 | 36.6 |
| | Completely to end or stop the pain (you could not go on living with the pain or how you were feeling) | 63 | 21.4 |

*Note: n*, Yes response of the aspects of ideation.

## Discussion

This epidemiological study documented the prevalence, severity and intensity of suicidal ideation and behaviors among youth aged 14–25 in Nairobi Metropolitan areas in Kenya. The key findings include a 19.9% lifetime prevalence of suicidal ideation, with higher proportions in peri-urban areas. Significant predictors included gender, education, marital status and economic status. These findings highlight multifactorial intervention points for suicide prevention. We provide data for future comparisons in different sites and future studies.

## Social demographic and economic factors

The higher prevalence of females was due to more women in nursing programs. The study, conducted during holidays, included both in-school and out-of-school youth. College students' ages vary, with an upper outlier of 25 years. Similar observations have been made in our previous studies (Ndetei et al., 2022). Additionally, the majority of the participants (15.30 + 37.8 = 53.1) were low-income, based on housing, toilets and energy; areas were selected with local administration in poor Nairobi and peri-urban settlements.

**Table 4.** The comparison of lifetime suicidal ideations, behavior (attempt and preparatory acts) across social demographics

| Variable | Category | Lifetime suicidal ideations | | Attempts | | Lifetime actual attempt | | Interrupted attempt | | Aborted or self-interrupted attempt | | Preparatory acts or behavior | |
|---|---|---|---|---|---|---|---|---|---|---|---|---|---|
| | | N (%) | $\chi^2$ (df); $p^*$ | N (%) | $\chi^2$ (df); $p^*$ | N (%) | $\chi^2$ (df); $p^*$ | N (%) | $\chi^2$ (df); $p^*$ | N (%) | $\chi^2$ (df); $p^*$ | N (%) | $\chi^2$ (df); $p^*$ |
| Gender | Female | 229 (21.6%) | $\chi^2$ (1) = 6.625; **p = 0.010\*** | 187 (18.0%) | $\chi^2$ (1) = 0; p = 0.99 | 150 (14.5%) | $\chi^2$ (1) = 0.264; p = 0.607 | 65 (7.0%) | $\chi^2$ (1) = 3.045; p = 0.081 | 62 (6.7%) | $\chi^2$ (1) = 0.258; p = 0.612 | 49 (5.3%) | $\chi^2$ (1) = 0.137; p = 0.711 |
| | Male | 147 (16.9%) | | 153 (18.0%) | | 130 (15.4%) | | 38 (5.0%) | | 46 (6.1%) | | 43 (5.7%) | |
| Location setting | Peri-urban | 33 (19.9%) | $\chi^2$ (1) = 0.006; p = 0.939 | 29 (17.7%) | $\chi^2$ (1) = 0.016; p = 0.899 | 25 (15.4%) | $\chi^2$ (1) = 0.027; p = 0.87 | 11 (7.5%) | $\chi^2$ (1) = 0.593; p = 0.441 | 5 (3.5%) | $\chi^2$ (1) = 2.205; p = 0.138 | 6 (4.2%) | $\chi^2$ (1) = 0.52; p = 0.471 |
| | Urban | 351 (19.6%) | | 317 (18.1%) | | 260 (15.0%) | | 93 (5.9%) | | 104 (6.6%) | | 87 (5.6%) | |
| Wealth Index | 1 | 71 (25.3%) | $\chi^2$ (4) = 8.908; p = 0.063 | 68 (25.1%) | $\chi^2$ (4) = 12.194; **p = 0.016\*** | 54 (20.1%) | $\chi^2$ (4) = 7.669; p = 0.105 | 20 (8.8%) | $\chi^2$ (4) = 6.054; p = 0.195 | 23 (9.9%) | $\chi^2$ (4) = 7.055; p = 0.133 | 27 (11.7%) | $\chi^2$ (4) = 19.377; **p = 0.001\*** |
| | 2 | 139 (20.0%) | | 117 (17.1%) | | 96 (14.2%) | | 41 (6.8%) | | 39 (6.5%) | | 28 (4.7%) | |
| | 3 | 130 (19.3%) | | 108 (16.3%) | | 92 (13.9%) | | 28 (4.7%) | | 30 (5.0%) | | 25 (4.2%) | |
| | 4 | 26 (15.9%) | | 34 (20.9%) | | 28 (17.5%) | | 12 (7.7%) | | 11 (7.3%) | | 9 (6.0%) | |
| | 5 | 2 (8.3%) | | 6 (25.0%) | | 5 (21.7%) | | 2 (9.1%) | | 1 (4.3%) | | 2 (8.7%) | |
| Age group | <18 | 57 (24.1%) | $\chi^2$ (1) = 3.349; p = 0.067 | 44 (19.0%) | $\chi^2$ (1) = 0.188; p = 0.665 | 25 (11.1%) | $\chi^2$ (1) = 3.018; p = 0.082 | 17 (9.0%) | $\chi^2$ (1) = 3.373; p = 0.066 | 25 (12.9%) | $\chi^2$ (1) = 14.98; **p = 0.000\*** | 16 (8.6%) | $\chi^2$ (1) = 3.529; p = 0.06 |
| | ≥18 | 317 (19.0%) | | 293 (17.9%) | | 253 (15.5%) | | 84 (5.7%) | | 83 (5.6%) | | 76 (5.2%) | |
| Religion | Protestant | 175 (21.7%) | $\chi^2$ (3) = 5.354; p = 0.148 | 143 (18.0%) | $\chi^2$ (3) = 1.666; p = 0.645 | 113 (14.4%) | $\chi^2$ (3) = 2.65; p = 0.449 | 52 (7.3%) | $\chi^2$ (3) = 4.762; p = 0.19 | 48 (6.8%) | $\chi^2$ (3) = 0.462; p = 0.927 | 40 (5.7%) | $\chi^2$ (3) = 3.318; p = 0.345 |
| | Catholic | 137 (17.1%) | | 151 (19.2%) | | 131 (16.8%) | | 34 (4.8%) | | 44 (6.2%) | | 33 (4.7%) | |
| | Muslim | 21 (20.6%) | | 14 (14.3%) | | 12 (12.4%) | | 6 (7.0%) | | 6 (6.8%) | | 8 (9.3%) | |
| | Other | 26 (19.4%) | | 22 (17.1%) | | 18 (14.2%) | | 5 (4.2%) | | 6 (5.4%) | | 6 (5.5%) | |
| Employment status | Unemployed | 287 (18.9%) | $\chi^2$ (3) = 3.054; p = 0.383 | 258 (17.3%) | $\chi^2$ (3) = 7.374; p = 0.061 | 210 (14.3%) | $\chi^2$ (3) = 4.002; p = 0.261 | 77 (5.8%) | $\chi^2$ (3) = 1.785; p = 0.618 | 83 (6.3%) | $\chi^2$ (3) = 8.059; **p = 0.045\*** | 77 (5.9%) | $\chi^2$ (3) = 5.247; p = 0.155 |
| | Volunteering | 52 (23.4%) | | 41 (18.6%) | | 38 (17.2%) | | 13 (6.5%) | | 12 (5.9%) | | 6 (3.0%) | |
| | Full time employment | 17 (23.0%) | | 21 (30.0%) | | 15 (21.4%) | | 6 (9.7%) | | 9 (14.1%) | | 6 (9.4%) | |
| | Part time employment | 26 (19.5%) | | 25 (18.9%) | | 22 (16.7%) | | 8 (7.0%) | | 4 (3.5%) | | 4 (3.7%) | |
| Highest level of education | Primary | 47 (26.9%) | $\chi^2$ (3) = 6.394; p = 0.094 | 48 (29.3%) | $\chi^2$ (3) = 16.055; **p = 0.001\*** | 30 (18.8%) | $\chi^2$ (3) = 2.614; p = 0.455 | 13 (9.0%) | $\chi^2$ (3) = 3.286; p = 0.35 | 24 (16.8%) | $\chi^2$ (3) = 28.912; **p = 0.000\*** | 20 (14.5%) | $\chi^2$ (3) = 23.335; **p = 0.000\*** |
| | Secondary | 191 (18.7%) | | 171 (17.1%) | | 147 (14.8%) | | 52 (5.9%) | | 48 (5.4%) | | 41 (4.7%) | |
| | Tertiary | 120 (19.4%) | | 107 (17.4%) | | 91 (14.9%) | | 34 (6.1%) | | 32 (5.8%) | | 25 (4.6%) | |
| | University | 26 (19.5%) | | 19 (14.3%) | | 16 (12.2%) | | 5 (3.9%) | | 5 (3.9%) | | 7 (5.5%) | |
| Current marital status | Not married | 312 (18.9%) | $\chi^2$ (2) = 7.349; **p = 0.025\*** | 281 (17.2%) | $\chi^2$ (2) = 18.989; **p = 0.000\*** | 235 (14.6%) | $\chi^2$ (2) = 12.306; **p = 0.002\*** | 84 (5.8%) | | 83 (5.7%) | | 70 (4.9%) | |
| | Previously married | 20 (32.8%) | | 23 (39.0%) | | 18 (30.5%) | | 8 (15.7%) | $\chi^2$ (2) = 8.579; **p = 0.014\*** | 10 (19.6%) | $\chi^2$ (2) = 16.648; **p = 0.000\*** | 12 (24.0%) | $\chi^2$ (2) = 35.282; **p = 0.000\*** |
| | Married | 45 (20.1%) | | 34 (15.9%) | | 27 (12.7%) | | 11 (5.6%) | | 11 (5.6%) | | 8 (4.2%) | |

(Continued)

**Table 4.** (*Continued*)

| Variable | Category | Lifetime suicidal ideations | | Attempts | | Lifetime actual attempt | | Interrupted attempt | | Aborted or self-interrupted attempt | | Preparatory acts or behavior | |
|---|---|---|---|---|---|---|---|---|---|---|---|---|---|
| | | N (%) | $\chi^2$ (df); p* | N (%) | $\chi^2$ (df); p* | N (%) | $\chi^2$ (df); p* | N (%) | $\chi^2$ (df); p* | N (%) | $\chi^2$ (df); p* | N (%) | $\chi^2$ (df); p* |
| Living situations | Lives alone | 79 (18.6%) | $\chi^2$(3) = 0.885; p = 0.829 | 73 (17.5%) | $\chi^2$(3) = 0.191; p = 0.979 | 70 (16.7%) | $\chi^2$(3) = 2.028; p = 0.567 | 17 (4.5%) | $\chi^2$(3) = 3.121; p = 0.373 | 12 (3.2%) | **$\chi^2$(3) = 11.632; p = 0.009***  | 22 (6.0%) | $\chi^2$(3) = 1.571; p = 0.666 |
| | Lives with immediate family | 244 (20.4%) | | 212 (18.1%) | | 169 (14.6%) | | 69 (6.7%) | | 79 (7.6%) | | 56 (5.5%) | |
| | Lives with partner/ relatives | 43 (19.7%) | | 38 (17.5%) | | 9 (13.6%) | | 14 (7.0%) | | 9 (4.6%) | | 8 (4.1%) | |
| | Lives with non-relatives/others | 15 (17.6%) | | 16 (19.0%) | | 10 (12.0%) | | 3 (3.9%) | | 8 (10.0%) | | 6 (7.7%) | |
| Place of abode/ residence | Owned or family-owned | 112 (21.1%) | $\chi^2$(2) = 0.938; p = 0.626 | 114 (22.1%) | **$\chi^2$(2) = 12.448; p = 0.002*** | 88 (17.3%) | **$\chi^2$(2) = 8.483; p = 0.014*** | 31 (6.8%) | $\chi^2$(2) = 2.096; p = 0.351 | 44 (9.6%) | **$\chi^2$(2) = 13.549; p = 0.001*** | 37 (8.2%) | **$\chi^2$(2) = 9.132; p = 0.010*** |
| | Hosted (relative or non-relative) | 21 (18.3%) | | 10 (9.0%) | | 7 (6.4%) | | 3 (3.0%) | | 2 (1.9%) | | 3 (2.9%) | |
| | Rented or institutional housing | 251 (19.3%) | | 221 (17.3%) | | 190 (14.9%) | | 70 (6.1%) | | 62 (5.4%) | | 53 (4.7%) | |

*Note:* Significant *p*-values are indicated in bold. *The chi-square statistic is significant at the .05 level. Attempts include actual, interrupted and aborted or self-interrupted attempts.

**Table 5.** Demographics as predictors of suicidal ideations, attempts and preparatory acts or behaviors (Lifetime)

| Variable | Category | Suicide ideations COR (95% CI) | Attempts COR (95% CI) | Actual attempts COR (95% CI) | Interrupted COR (95% CI) | Aborted or self interrupted attempt COR (95% CI) | Preparatory acts or behavior COR (95% CI) |
|---|---|---|---|---|---|---|---|
| Gender | Female | REF | _ | _ | REF | _ | _ |
| | Male | **0.76 (0.59–0.98) *** | _ | _ | **0.63 (0.40–0.998) *** | _ | _ |
| Age | | 0.98 (0.93–1.03) | _ | **1.07 (1.01–1.13) *** | 0.95 (0.87–1.03) | 0.97 (0.89–1.05) | 1 (0.92–1.1) |
| Wealth index | 1 Lowest | REF | REF | REF | REF | REF | REF |
| | 2 | 0.81 (0.56–1.17) | 0.73 (0.51–1.05) | 0.73 (0.49–1.07) | 0.81 (0.44–1.5) | 0.84 (0.45–1.55) | **0.44 (0.24–0.8) **** |
| | 3 | 0.79 (0.55–1.15) | 0.7 (0.49–1.02) | 0.74 (0.5–1.1) | 0.57 (0.29–1.09) | 0.65 (0.34–1.23) | **0.38 (0.2–0.71) **** |
| | 4 | 0.58 (0.34–1.01) | 0.9 (0.55–1.48) | 0.92 (0.54–1.57) | 0.8 (0.35–1.85) | 0.94 (0.4–2.18) | 0.54 (0.23–1.27) |
| | 5 Highest | 0.32 (0.07–1.42) | 1.05 (0.37–3.01) | 1.17 (0.41–3.34) | 1.23 (0.26–5.84) | 0.69 (0.08–5.73) | 0.83 (0.18–3.94) |
| Religion | Protestant | REF | _ | _ | REF | _ | _ |
| | Catholic | **0.74 (0.57–0.96) *** | _ | _ | 0.68 (0.43–1.08) | _ | _ |
| | Muslim | 0.84 (0.48–1.48) | _ | _ | 0.57 (0.17–1.89) | _ | _ |
| | Other | 0.82 (0.51–1.34) | _ | _ | 0.54 (0.21–1.41) | _ | _ |
| Marital status | Not Married | REF | REF | REF | REF | REF | REF |
| | Previously Married | 1.63 (0.87–3.05) | **2.65 (1.51–4.67) ***** | **2.2 (1.21–4.03) *** | 1.85 (0.69–4.94) | **3.48 (1.54–7.88) ***** | **4.62 (2.14–9.98) ******* |
| | Married | 1.01 (0.67–1.51) | 0.87 (0.58–1.31) | 0.75 (0.48–1.19) | 1.06 (0.54–2.11) | 1.07 (0.45–2.52) | 0.78 (0.35–1.74) |
| Level of education | Primary | REF | REF | _ | _ | REF | REF |
| | Secondary | **0.65 (0.43–0.997) *** | **0.58 (0.38–0.87) ***** | _ | _ | **0.39 (0.21–0.74) ***** | **0.43 (0.23–0.83) *** |
| | Tertiary | 0.75 (0.47–1.19) | **0.62 (0.4–0.97) *** | _ | _ | **0.47 (0.22–0.99) *** | **0.42 (0.19–0.89) *** |
| | University | 0.89 (0.47–1.66) | **0.52 (0.27–0.99) *** | _ | _ | 0.37 (0.12–1.12) | 0.67 (0.24–1.84) |
| Employment Status | Unemployed | | REF | _ | _ | REF | REF |
| | Volunteering | | 1.13 (0.77–1.67) | _ | _ | 1.21 (0.63–2.33) | 0.5 (0.21–1.2) |
| | Full time employment | | 1.73 (0.97–3.09) | _ | _ | 2.26 (0.93–5.53) | 0.84 (0.28–2.59) |
| | Part time employment | | 1.24 (0.77–1.99) | _ | _ | 0.7 (0.24–2.03) | 0.64 (0.22–1.87) |
| Location setting | Peri-urban | | _ | _ | _ | 2.16 (0.7–6.66) | _ |
| | Urban | | _ | _ | _ | _ | _ |

(*Continued*)

**Table 5.** (*Continued*)

| Variable | Category | Suicide ideations COR (95% CI) | Attempts COR (95% CI) | Actual attempts COR (95% CI) | Interrupted COR (95% CI) | Aborted or self interrupted attempt COR (95% CI) | Preparatory acts or behavior COR (95% CI) |
|---|---|---|---|---|---|---|---|
| Living situations(Lives with) | Alone | _ | _ | _ | | REF | _ |
| | Immediate family | _ | _ | _ | | 1.76 (0.88–3.54) | _ |
| | Partner/Relatives | _ | _ | _ | | 1.11 (0.37–3.29) | _ |
| | Non-relatives/Others | _ | _ | _ | | **5.46 (2–14.92) \*\*** | _ |
| Place of abode/Residence | Owned or family-owned | REF | REF | _ | | REF | REF |
| | Hosted (relative or non-relative) | **0.38 (0.19–0.76) \*\*** | **0.29 (0.12–0.69) \*\*** | _ | | **0.19 (0.04–0.85) \*** | 0.41 (0.12–1.38) |
| | Rented or institutional housing | 0.79 (0.6–1.03) | 0.85 (0.63–1.14) | _ | | 0.71 (0.44–1.14) | 0.66 (0.41–1.07) |

*Note:* REF is the reference category. AOR, adjusted odds ratio, CI, confidence interval at 95%; COR, Crude odds ratio. The bolded COR and AOR at 95% CI are significant at $p < 0.05$. The regression analysis result \*, \*\* and \*\*\* is significant at $p < .05$, $p < 0.01$ and $p < 0.001$ level, respectively. Suicidal ideations, attempts and preparatory acts were the dependent variables treated as binary (yes/no). The model predicts the odds of a yes response for each outcome. Some of the confidence intervals are shown to three decimal places to avoid rounding errors near the significance threshold.

### Suicidal ideation and behavior

**Frequency of ideation (lifetime and recent-past 1 month):** There is no total agreement with previous studies in actual prevalence as detailed in the Introduction, but the trends remain similar. Suicidal ideation was more frequent than suicidal behavior. About 5.4% lifetime and 2.7% recent did not intend to kill themselves when they made the suicidal attempts (item #13 in Table 2), but a slightly higher prevalence (#9, 10 and 11 in Table 2) wanted to die. While these frequencies are lower compared to many other reports in Africa, these prevalences should be taken seriously, given that 3 out of 100 attempted suicides intended to die, 14.2% had recent intent and 11.7% had a specific plan and intent.

Significant differences were observed between urban and peri-urban areas, with peri-urban participants showing a higher prevalence of suicidal ideation but lower prevalence of self-harming behavior without suicidal intent and fewer instances of stopping themselves before attempting suicide compared to their urban counterparts, a pattern that contrasts with previous studies suggesting higher mental health vulnerability in peri-urban settings due to limited access to services (Silva et al., 2016).

*Intensity of ideation*: Our findings suggest that recent suicidal ideas have a very high intensity, occurring almost daily and lasting several hours a day. This specific finding has been demonstrated in a Kenyan study that suicidal attempts at the time they are made can occur without preceding ideas/thoughts or expressed (Ndetei et al., 2024). The intensity of suicidal thoughts surpasses deterrent factors, which stresses the requirement for assessment using the format in Table 2 as the standard. The finding that at least 25% of individuals had suicidal intent during their entire lifetime while 26% of active cases present thoughts about taking their life underlines the need to proactively ask for these in the youth.

*Reasons for suicidal ideations (drawn from open-ended responses among participants who endorsed such thoughts)*: Depression was the most stated reason for suicidal ideation, but other factors such as family conflict, loneliness, discrimination, bullying and financial difficulties play a crucial role and should be targeted in the intervention, confirming earlier Kenyan findings (Musyimi et al., 2020; Mutiso et al., 2019; Mutwiri et al., 2023; Ndetei et al., 2024).

*The prevalence of suicidal behavior (lifetime and recent-past 3 months)*: Our findings reveal a broader spectrum of suicidal behavior than previously reported in Kenya, notably that non-suicidal self-injuries emerged as the second most common suicidal behavior, despite individuals recognizing the potential lethality of their actions. Seeking attention or help was the main motive. Our results confirm the long-standing findings that lifetime attempts are more than in the last 3 months, suggesting chronic suicidality. Suicide attempts occurred in 6.7% (lifetime) and 3.6% (last 3 months), thwarted by others (6.4%, 4%) or self-stopped (6.4%, 4%) in Kenya. Protective factors, as discussed below, likely played a role in halting these suicidal behaviors before they were carried out.

*Activating events:* The finding that 29% of those who had suicidal behavior had a recent negative life event has been found in other studies (Mortier et al., 2021; Musyimi et al., 2020), thus underscoring the importance of exploring significant life events during clinical evaluations of suicidal behavior.

*Psychiatric history*: A third (33.3%) of suicidal patients received no treatment; 23.4% relapsed due to non-compliance (20%) or dissatisfaction (19.8%), highlighting the need to involve caregivers, finding workable solutions or alternatives and monitoring compliance. Dissatisfaction was based on participant self-report and often reflected perceived ineffectiveness, medication side effects or lack of follow-up care. These findings underscore the importance of improving therapeutic communication, follow-up systems and offering alternative or youth-friendly treatment options to increase engagement and reduce dropout (Beckwith, 2021; Tenhovuori, 2024).

*Protective factors*: The family is one of the most important protective factors, with psychosocial support from family and others playing a significant role. Engagement in activities such as work and school also serves as a protective factor. In Kenya, we have found this to be an effective treatment for dealing with depression and anxiety in youth (Ndetei et al., 2008; Osborn et al., 2023). Spiritual beliefs can help in coping with mental illness, but there is a need to know it is a medical condition and not supernatural, thus the need to educate the clergy, the traditional healers and the parents to understand and be aware of the role supernatural power has in coping. The parents of the youth should also be part of this awareness.

*Types of suicidal behavior*: Unlike earlier studies where hanging was prevalent, our urban/peri-urban study found reckless pedestrian road behavior as the most common suicide attempt. Environmental factors like high traffic, overcrowding and mob justice may also contribute. Protective measures on high-rise buildings and flyovers remain crucial to prevention, as some may provoke mob violence seeking death (Hemmer et al., 2017; Marzec et al., 2021). Suicidality is influenced by various environmental factors from location to period; therefore, interventions need to adapt to these trends.

### Associations and predictors of suicidal ideations and behavior

Females showed a higher prevalence of suicidal ideations due to greater psychological distress, aligning with other research findings (Griffin et al., 2022). Males had lower odds of interrupted suicide attempts (COR = 0.63, 95% CI: 0.4–1) (Cibis et al., 2012). Younger respondents were more prone to interrupted suicide attempts, reflecting impulsivity and the development of emotional regulation (Gifuni et al., 2020). Older adults were more likely to have actual attempts, aligning with research on age-related suicidality vulnerabilities (Conejero et al., 2018).

Marital status had a significant association with suicidal ideations and behavior. While marriage is more common in older youth, we included the entire age group (14–25 years) in the analysis to reflect the contextual reality of early marriages and informal unions, which are not uncommon in low-resource settings, including urban informal settlements in Kenya. Participants who were married had greater thoughts of suicide, but individuals who have never been married tend to carry out suicide attempts and prepare for them potentially because they lack support systems, which can result in helpless feelings and dangerous behaviors. In contrast, marriage acts as a protection against suicide because it gives individuals access to emotional and social backing (McLean et al., 2008). The process of divorce or separation tends to increase suicide risks because loneliness, financial troubles and decreased social support increase psychological problems (Kołodziej-Zaleska & Przybyła-Basista, 2016).

Education level emerged as a protective factor against suicidal ideations and behaviors, with higher educational attainment reducing suicide potential through better coping abilities and improved self-recognition together with service access (Mutwiri et al., 2023). Preparatory acts occur more frequently within the primary education level. Place of residence also played a role, with people who live in hosted accommodations like those with family or non-family members, showing significantly decreased prevalence of suicidal

ideations and attempts compared to residents of family-owned homes, which indicates protective social support (McLean et al., 2008; Stansfeld et al., 2006). Economic status as measured by the wealth index did not reliably predict suicidal ideations or attempts. People with lower levels of wealth experience a greater risk of attempting suicide and engaging in deadly preparations, but poverty serves to intensify suicide-related activities (Mutwiri et al., 2023; Ndetei et al., 2022). The family dynamically influences suicidal tendencies exhibited by Kenyan youth. The goal of awareness campaigns is to support Kenyan families while training beneficiaries to practice harmonious mutual support along with understanding suicide behaviors relevant to Kenyan society.

### Limitations and recommendations

The cross-sectional design of our study limits the ability to establish causal relationships between variables and provides only a snapshot of suicidality at a single point in time. Future studies should consider adopting a longitudinal design to better understand the long-term trends in suicidality, allowing for a deeper exploration of the factors that contribute to changes over time. We acknowledge that the selection of informal settlements as part of the study sites may introduce some bias and limit generalizability to wealthier urban or rural youth populations. However, this was mitigated by recruiting participants from peri-urban regions and educational institutions to ensure a broader representation. As such, the findings have reasonable external validity and may be cautiously generalized to youth in similar low-resource urban settings in Kenya and comparable contexts.

### Conclusions

Our study advances real-time identification of suicide attempts in Kenya, uncovering undocumented patterns of suicidality. This study has confirmed that there are high levels of suicidal ideation and behavior among youth in Nairobi, particularly among peri-urban populations. Clinical factors such as ideation severity and behavioral intensity were significantly associated with social and economic status. The high prevalence in non-clinical populations suggests similar urban and low-income neighborhoods.

We have achieved our general objective and specific aims.

**Open peer review.** To view the open peer review materials for this article, please visit http://doi.org/10.1017/gmh.2026.10148.

**Supplementary material.** The supplementary material for this article can be found at http://doi.org/10.1017/gmh.2026.10148.

**Data availability statement.** Requests for the data may be sent to the corresponding author.

**Author contribution.** Conceptualization: D.N., D.M.; Critique of the manuscript: D.W., K.B., J.S., S.W., T.L.O., M.S., A.S., D.M.; Drafting of the paper: D.N.; Draft review: P.N.; Field work during data collection and literature review: V.O.; Literature review: E.J.; Oversight of data collection: V.M.; Oversight on ethics: C.M.; Statistical analysis: S.W., E.J. All authors read and approved the final manuscript.

**Financial support.** This study was funded by National Institutes of Health (NIH), Grant/Award number: 5R01MH127571–02.

**Competing interest.** The authors declare no conflict of interest.

**Ethics statement.** The authors assert that all procedures contributing to this work comply with the ethical standards of the relevant national and institutional committees on human experimentation and with the Helsinki Declaration of 1975, as revised in 2008. All procedures involving human subjects/patients were approved by the Nairobi Hospital Ethics Research Committee (approval no. TNH-ERC/DMSR/ERP/022/22). The study obtained licensing from the National Commission for Science, Technology and Innovation (NACOSTI) license number NACOSTI/P/22/18097. Permissions were obtained from county offices and colleges. Before data collection, adults provided informed consent while minors under 18 provided assent accompanied by a parent or legal guardian who provided consent in line with ethics committee guidance. Consent procedures were conducted in private spaces to ensure voluntary participation and confidentiality. Participants determined to be at risk of suicide were referred to public psychiatric facilities for evaluation on underlying psychiatric conditions and care. Participants were given verbal study explanations, could ask questions and could withdraw at any time without penalty.

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
