## [Reviewer Report]

1. The title lacks clarity. Please pay attention to the following in the title: Various Clinical and Non-Clinical Associates and Indicators (“Various Clinical and Non-Clinical Associates and Indicators”); Repetitive or abstract terms (“Associates and Indicators”).

2. This study aims to push the boundaries of indicators of suicidality and hence enhance known entry points for intervention in the study context. What specific “boundaries” of suicidality indicators are you referring to, and how does your study aim to push them.

3. The general objective of this study is to fill the above-specified gap among youth in a specific environment in the Nairobi metropolitan area. Consider revising the sentence to past tense to reflect that the study has already been completed.

4. Since the study appears to be completed, it would be clearer and more appropriate to put the objectives in the past tense. Also, consider making the specific aims more concise and parallel in structure. For example, aim 2 could be rephrased for clarity: “To investigate clinical features of suicidality—including ideation, behavior, attempts, severity, and intensity—that may inform potential interventions. Aim 3 could specify what is meant by ”environmental non-clinical factors" to improve clarity.

5. The inclusion of both in-school and out-of-school youths is a strength of the study. However, it would be helpful if the authors explicitly discussed the rationale for involving these two groups. Specifically, clarifying how the experiences or risk factors for suicidality might differ between in-school and out-of-school youths, and how this diversity informs the study design and potential interventions.

6. The authors state that participants aged 14–25 years were enrolled and that ‘both adults and minors’ provided written consent. While the age categorization is clear, the consent process for minors (under 18) needs clarification. Specifically, did the study obtain parental or guardian consent in addition to assent from minors, or was a waiver of parental consent approved by the ethics committee? This clarification is important for assessing the ethical rigor of the study.

7. Given that participants were gathered in a common setting (e.g., schools, or community centers), it would be important for the authors to describe how the consent and assent processes were conducted. Specifically, how was privacy ensured during consent, how was voluntary participation emphasized, and how were procedures differentiated for minors versus adults? Clarifying these aspects would strengthen the ethical transparency of the study.

8. The authors report that none of the participants declined to participate, noting a near 100% response rate consistent with previous studies. However, to fully interpret this finding, it is important to provide details on how many individuals were initially approached or screened for eligibility, and how many were excluded due to ineligibility. This information is critical to assess potential selection bias and to understand the representativeness of the final sample.

9. The authors write “With standardized definitions and questions, the C-SSRS improves the accuracy of suicide risk assessment (Yershova et al., 2016) and has changed the consideration of suicide among practice, research, and public health (Posner et al., 2014)” (page 9, lines 31-37). This sentence makes important points about the value of the C-SSRS; however, the phrase ‘changed the consideration of suicide’ is vague and could be reworded to more precisely convey the scale or nature of its impact. Additionally, separating the statements about accuracy (Yershova et al., 2016) and influence on practice, research, and public health (Posner et al., 2014) may improve clarity and strengthen the logical flow.

10. The authors state that the study aimed to investigate environmental non-clinical factors (objective 3) that may inform intervention. However, the variables assessed which are limited to sociodemographic characteristics and wealth index may not sufficiently reflect the broader range of environmental factors typically considered relevant to suicidality (e.g., family environment, peer influence, social support, community context). The authors should clarify the rationale for defining environmental factors narrowly, or consider revising the objective to more accurately align with the variables assessed.

11. The C-SSRS is a self-administered tool that measures suicide ideation and behavior. The first scale (Suicidal ideation) assesses 5 levels of ideation severity, ranging from 1 (wish to be dead) to 5 (suicidal intent with plan). You probably meant “the first sub scale”. Consider revising.

12. The authors indicate that data were collected using a questionnaire, yet some results, such as ‘Reasons for Suicidal Ideations,’ are described as narrative, although they are reported as percentages. Could the authors clarify whether this item was open-ended or closed-ended, and explain how narrative responses were quantified? If qualitative responses were thematically analyzed, please describe the analytic process used.

13. The opening paragraph of the discussion reiterates the study’s objectives and future relevance but does not summarize the main findings. To improve clarity and coherence, the authors should consider briefly highlighting the key results — such as the most significant associations between clinical features, environmental variables, and suicidality — before delving into interpretation and implications. This would help anchor the discussion and guide the reader through the authors’ line of reasoning.

14. The authors note that both in-school and out-of-school youths were included, yet the methods section does not clearly define what constitutes ‘out-of-school’ youth (e.g., school dropouts, graduates, unemployed, or employed youth). Additionally, the proportions of in-school versus out-of-school participants are not reported, which limits the reader’s ability to interpret subgroup differences or assess representativeness. The authors should clarify their definitions and provide a breakdown of participant status.

15. The authors state that a majority of participants (53.1%) were low-income, based on housing, toilets, and energy indicators, and that areas were selected in collaboration with local administration in poor Nairobi and peri-urban settlements. However, the criteria for defining ‘low-income’ are not clearly described, and it is unclear how these indicators were operationalized. Furthermore, the process for site selection in collaboration with local authorities needs more detail to assess potential sampling bias. The authors should clarify both the definition and measurement of socioeconomic status, as well as the rationale and method used for site selection.

16. Psychiatric history: A third (33.3%) of suicidal patients received no treatment; 23.4% relapsed due to non-compliance (20%) or dissatisfaction (19.8%), highlighting the need to involve caregivers, finding workable solutions or alternatives and monitoring compliance (page 20, lines 28-32). This section presents important findings on treatment gaps among suicidal patients, including non-treatment, relapse, and non-compliance. However, the link between these findings and the proposed solutions (caregiver involvement, finding alternatives, and monitoring compliance) would benefit from clearer explanation. For example, how was dissatisfaction with treatment assessed? What specific types of support or alternatives are being suggested? Expanding on these points would enhance the interpretation and make the recommendations more actionable.

17. The conclusion summarizes broad implications of the study, such as advancing real-time identification of suicide attempts and highlighting clinical and public health relevance. However, it lacks specificity about the key findings that support these claims.

18. The conclusion in the main manuscript differs in content from the one presented in the abstract. For clarity and consistency, the authors should ensure alignment between the abstract and full-text conclusions, and explicitly summarize the main findings that led them to these conclusions.

---

## [Reviewer Report]

Congratulations on selecting this relevant topic.

Kindly address the following concerns to improve the manuscript

In the results section of the abstract, it is mentioned that ‘Our findings re-confirm the high produce of suicidal ideas and behavior in the youth (19.9% and 3.6%)’. It would be good to replace the word ‘produce’ with’ prevalence’ or’ proportion’. Kindly mention the sampling technique in the abstract.

In the objective section, it is mentioned that ‘The general objective of this study is to fill the above-specified gap among youth in a specific environment in the Nairobi metropolitan area.’ Kindly reframe the objective to make it self-explanatory and stand alone. The same is applicable to other specific objectives. Alternatively, these can be stated as primary and secondary objectives.

Kindly mention the justification for the sample size and describe the sampling technique. Kindly mention the external validity of the results.

Kindly mention whether the presence of depression was studied, especially among those with strong suicidal ideas and attempts. Is there any data available on the proportion of referred persons who sought care from a mental health professional?

All the tables present a comparison of the urban and per-urban subsamples. This comparison is not explicitly stated in the title or objective. Ideally, the analysis on determinants should categorise the outcome into those with suicidal ideas and those without suicidal ideas. Kindly provide a re-analysis.

Kindly provide a baseline table first.

Kindly clarify whether there is any scoring system available to compare the total score of the tool and, mean difference between groups?

There are items (for example, items 6,7,8 etc) which elicit life time experience/exposure (item 6- Have you made a suicide attempt in your lifetime?) . In table 2, the responses have been categorized into lifetime and recent. Did you alter the items or did you use another questionnaire to another questionnaire to collect data on the recent experience. In the section on tools, it is mentioned that ‘The self-reported full version of the C-SSRS assessed lifetime and recent (1 month) suicidality’. But these are not listed in the table 2. Kindly clarify. If not, kindly modify this table also.

Kindly explain the methodology for data collection, on the narratives (causative factors, protective factors). Did you use qualitative techniques? Kindly clarify this in the methods section and address the same in the results and discussion.

The entire data { age goup (14-25)} is used in the analysis of marriage as a protective factor. It would be more appropriate to compare adults who are married and unmarried. Otherwise, justify the inclusion of all age groups in the analysis. (Is child marriage common?).

Kindly add a brief note on the scoring system C-SSRS and analysis

Finally, kindly do an editing for English language and grammar to improve the reading and comprehending quality of the manuscript

Once again congratulations for choosing to work in the area of suicide prevention. Best wishes

---

## [Reviewer Report]

I do not think this manuscript is ready for publication. I would recommend that it be reviewed by a mentor before being resubmitted. Note, I’ve had to query nearly every other sentence for clarification. The writing needs to be drastically improved. At this state, I am unable to comment on the scientific merit of the manuscript. Please feel free to use my feedback below as you continue to develop this work.

Abstract

- Line 29: re-confirm should be “confirm”; produce is not the right word, unclear

- Line 33: sentence not clear

Impact statement

- Line 47 – delete “fundamental”

- General

o the study is documenting the prev. of different components of suicide (behaviors and thoughts) and examining differences by characteristics. It isn’t really touching on suicide rates.

o If you comment on the prevalence of suicidal thinking, you should give a time period (over the past XX amount of time).

o “Social demographics…” this sentence isn’t making a point

Introduction

- Sentence 1 is a run-on sentence that likely needs additional citations.

- Line 18: “high-income countries”

- Line 18: are thoughts behaviors? You likely want to define thoughts and behaviors up front. Confusing as written.

- Line 28 – 30: sentence not clea

- Line 32: 82% is really high. Is a referral hospital youth center specifically a mental health facility? Hard to contextualize as written

- Line 37: ideation over what period of time

- Paragraph two – again confusing how you are defining different aspects of suicidality.

- Line 3, page 6 “push the boundaries” sentence doesn’t make sense. I would delete this sentence.

- General Objective – delete heading. I think you should write ou what the objective is. At this point it’s not clear. I think (maybe) your objective is: “This study aims to document the severity and intensity of suicidal ideation and behaviors among youth age #-# in the Narobi metropolitan area and evaluate differences by relevant environmental risk factors that can be used to inform suicide prevention interventions.” Or something like that.

- Specific aims – note in a peer-reviewed manuscript, you would not include your specific aims, but summarize in a sentence or two. From example: Specifically, we aimed to 1) …, 2)… and 3)…

Methodology

Page 6

- Line 16 – I think you could revise “Two informal settlements areas located close to high-end city neighborhoods were chosen to ensure a diverse sample.” HOWEVER, if you’re only sampling poor neighborhoods, you’re likely NOT getting a diverse sample? Also, what does “high-end” mean?

- Line 30 – what does specifically in colleges mean, if they’re both in and out of school?

- Line 32 – the study aim doesn’t belong here

- Lines 47-55 – you would just describe the RAs and they’re training, not how you selected them in a peer-reviewed manuscript.

Page 7

- Line 8 – what did the sensitization phase include?

- Line 18 – who is “they” ?

- Line 28 – who are marginalized groups? Were specific efforts made to include them? Why did you highlight this?

- Line 37 – put the total population size in the results section, not the methods section

- Line 48 – Define C-SSRS at first use; Really not clear what the process for administering the survey was. Why were participants put in groups? Were the participants read the questions? Are the group merely used to schedule when participants should come in to do the survey?

Page 8

- Line 51 – what is classified into 5 sections? Don’t start sentences with “it.” Do you mean the self-report questionnaire yields 5 wealth categories?

Page 10

- Lines 9-13 – The C-SSRS is not usually operationalized as a score. Really not clear why and/or how you calculated cronbach’s alphas

- Lines 22-53 – It is not possible to follow what scales you used here. I’m familiar with the C-SSRS questions, but I can’t tell from your description how the C-SSRS was delivered. I also cannot follow your intensity sub-scales or behavioral scales. These would need citations or you should include the actual questions.

Page 12

- Line 6 – list-wise deletion of what?

- Really unclear from your methods what you planned to do

Discussion

General

- What was the “recent” time frame referring to?

---

## [Reviewer Report]

Well written. Congratulations

Long sentences may be rewritten as short and simple sentences. A single message in one sentence would be ideal for clarity.

In table 4, the title mentions about distribution only, while the table content presents data on comparison. Kindly modify the title appropriately.

In the results section, it is mentioned that ‘Nearly a quarter (23.4%) of the participants reported having a history of previous psychiatric diagnoses and treatments, while 18.8% reported feeling hopeless or dissatisfied with their treatment. Additionally, 19.8% were noncompliant with treatment, and 33.3% were not receiving treatment at all.”

Kindly clarify the denominator; whether it is those with suicidal thoughts or the total sample.

‘Males had a lower likelihood of suicidal ideation (COR = 0.76, 95% CI: 0.59–0.98) and interrupted attempts (COR = 0.63, 95% CI: 0.4–1), compared to females.’

In the above sentence, please note that the upper limit of 95% CI of crude odds ratio includes 1.

---

## [Editor Report]

Dear authors, 

Please consider the additional reviewer comments on your manuscript and address these comments in a minor revision. Additionally, in my own reading of the manuscript, I believe that the writing problems noted by reviewer 3 have been significantly improved; however, there are still a few areas with grammatical errors. Please review the manuscript thoroughly and correct these issues and those noted in this additional review before resubmitting the manuscript for consideration. Thank you.

---

## [Editor Report]

Dear Authors, 

We have reviewed your response to the second set of reviewer comments and believe that your manuscript is now ready for publication. Thank you for choosing to publish your work in the special issue on Self-harm and Suicide: A Global Priority in Global Mental Health. 

Kristin Kosyluk

Guest Editor